computational chemistry/physical chemistry

cobalt disulfide, density function theory, surface energy, thermodynamical stability, electronic structure

**Author for correspondence:**
Yan-li Zhu
e-mail: zhuyanli1999@bit.edu.cn

# Calculation on surface energy and electronic properties of CoS$_2$

Yan-li Zhu[1], Cong-Jie Wang[1], Fei Gao[2], Zhi-xia Xiao[1], Peng-long Zhao[3] and Jian-yong Wang[4]

[1]State Key Laboratory of Explosion Science and Technology, Beijing Institute of Technology, Beijing 100081, People's Republic of China
[2]Battery Energy Storage Technology Laboratory, China Electric Power Research Institute, Beijing 100192, People's Republic of China
[3]Qaidam Xinghua Lithium Salt Co., Ltd, No. 1 Dahua Street, Dachaidan, Haixi, Qinghai, 817000, People's Republic of China
[4]State Key Laboratory of Advanced Chemical Power Sources, Guizhou Meiling Power Sources Co. Ltd., Zunyi, Guizhou 563003, People's Republic of China

Y-lZ, 0000-0003-2581-2067

Density functional theory was employed to investigate the (111), (200), (210), (211) and (220) surfaces of CoS$_2$. The surface energies were calculated with a sulfur environment using first-principle-based thermodynamics. It is founded that surfaces with metal atoms at their outermost layer have higher energy. The stoichiometric (220) surface terminated by two layer of sulfur atoms is most stable under the sulfur-rich condition, while the non-stoichiometric (211) surface terminated by a layer of Co atoms has the lower energy under the sulfur-poor environment. The electric structure results show that the front valence electrons of (200) surface are active, indicating that there may be some active sites on this face. There is an energy gap between the stoichiometric (220) and (211), which has low Fermi energy, indicating that their electronic structures are dynamically stable. Spin-polarized bands are calculated on the stoichiometric surfaces, and these two (200) and (210) surfaces are predicted to be noticeably spin-polarized. The Bravais–Friedel–Donnay–Harker (BFDH) method is adopted to predict crystal growth habit. The results show that the most important crystal planes for the CoS$_2$ crystal growth are (111) and (200) planes, and the macroscopic morphology of CoS$_2$ crystal may be spherical, cubic, octahedral, prismatic or plate-shaped, which have been verified by experiments.

# 1. Introduction

Transition metal sulfides have been widely applied in various technological areas, such as optical devices, electrical, catalytic and

biological imaging because of their good electronic and magnetic properties [1–4]. Mitzi *et al.* [5] prepared the high-mobility ultrathin semiconducting films based on tin sulfide prepared by spin coating, which were used in thin-film field-effect transistors and exhibited large current densities and mobilities. Ramos's group confirmed that the complexes of cobalt sulfide and molybdenum sulfide have high activity and catalytic performance through the detailed characterization by X-ray diffraction (XRD), X-ray photoelectron spectroscopy (XPS) and high-resolution transmission electron microscopy (HRTEM) [6]. Our group obtained the highly uniform cobalt sulfide through a simple hydrothermal route, and it showed typical pseudocapacitive properties with a high specific capacitance and an excellent cycling stability [7].

Among them, $CoS_2$ is one of the ideal cathode materials in thermal batteries with both high power and high energy output capacity [8]. For recent years, there has been considerable interest in investigating the synthesis and characterization of $CoS_2$, and a number of different morphologies have been obtained, such as cubic, octahedron, and hollow microspheres [7,9,10]. There are also some reports on the theoretical calculation of $CoS_2$ [11,12]. Shishidou *et al.* compared the calculations on the half-metallicity of $CoS_2$ under generalized gradient approximation (GGA) and local spin density approximation (LSDA), which showed that the semi-metallic properties of $CoS_2$ calculated by the former are more obvious [13]. Wu *et al.* [14] predicted that the half-metallic gap might be controlled by antibonding S p rather than Co $e_g$ states by calculating the electronic band structure of $CoS_2$. They also discussed the spin bands of $CoS_2$ (001) by experiment and computation [15]. Liu & Altounian [16] studied the effect of pressure on the itinerant ferromagnet $CoS_2$ by first principles. While the theoretical calculations are mainly focused on the electronic structure and the magnet properties of $CoS_2$, few publications have reported the surface properties of $CoS_2$ crystal systematically, and the surface properties are directly related to the crystals morphology, which affects the macroscopic properties of the material.

Even though a lot of studies on surfaces of $FeS_2$ have been reported [17,18], as we know, sulfides which crystallize in pyrite structure ($FeS_2$, $CoS_2$, $NiS_2$, $CuS_2$ and $ZnS_2$) show great variation in electronic properties by changing transition metal ion or by substituting Se or Te in the anion site [19–21]. It is necessary to study the surface properties of $CoS_2$ systematically in order to better control the morphologies and the properties of crystal. In this paper, we investigated various terminations of the (111), (200), (210), (211) and (220) surfaces of $CoS_2$ using density functional theory (DFT). Both stoichiometric surfaces and non-stoichiometric surfaces were considered, and the variety of the sulfur chemical potential was also taken into account.

## 2. Computational approach

The standard powder diffraction pattern of $CoS_2$ in Joint Committee on Powder Diffraction Standards (JCPDS) showed that the major diffraction peaks can be ascribed to the (111), (200), (210), (211) and (220) surfaces of $CoS_2$, which were selected to be discussed in this paper.

The total energy calculations were carried out using first-principles spin-polarized DFT with the Cambridge Sequential Total Energy Package (CASTEP) program [22]. The spin-polarized effect was considered since the system was magnetic. The GGA formulation of Perdew, Burke and Ernzerhoff (PBE) were used to calculate the exchange-correlation energy [23]. The electron–ion interaction was described by the ultra-soft pseudo potential [24]. Broyden–Fletcher–Goldfarb–Shanno (BFGS) algorithm was employed to optimize the model geometry.

For the bulk cell of $CoS_2$ (containing four formula units of $CoS_2$), Brillouin-zone sampling was performed on a dense Monkhorst–Pack k-point mesh of $6 \times 6 \times 6$ points and cut-off energy was set as 450 eV. To simulate the various terminations of (111), (200), (210), (211) and (220) $CoS_2$ surfaces, we used slab technique with periodic boundary conditions imposed in the two directions parallel to the slab. To ensure the decoupling of the adjacent slabs, a 12 Å thick vacuum region along the surface normal was employed. The slab thickness was between 10 and 16 atomic layers based on the restriction of computational ability. All the atoms of the slab were relaxed with the eight bottom layers fixed to their bulk values. The lattice constants were fixed at bulk optimized conditions. For the various surface slab model, Brillouin-zone sampling was performed on a dense Monkhorst–Pack k-point mesh of $4 \times 4 \times 1$ points and cut-off energy was set as 280 eV. It was found that there was little change if a higher cut-off parameter was used (i.e. less than 5 meV Å$^{-2}$).

The stability of the various considered surfaces were investigated by associating DFT results with thermodynamic concepts [25–28]. The surface free energy in equilibrium with particle reservoirs at temperature $T$ and pressure $p$ is defined as

$$\gamma(T,p) = \frac{1}{A}\left[ G^{surf} - \sum_i N_i \mu_i(T,p) \right],$$ (2.1)

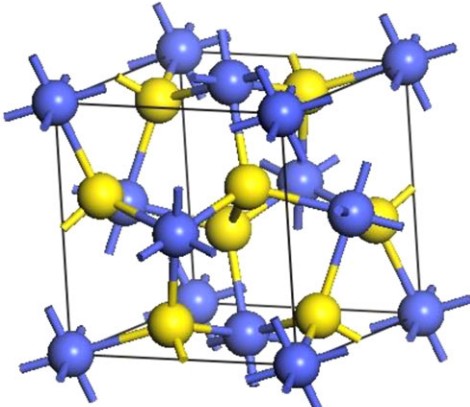

**Figure 1.** Bulk unit cell of CoS$_2$. Blue spheres indicate Co atoms and yellow spheres indicate S atoms.

Here, $G^{\mathrm{surf}}$ denotes the Gibbs free energy of a periodic repeated slab, which exposes a surface with area $A$. Since the two exposed faces of the slab are not symmetrically equivalent, the factor of $A$ is 1. The terms $N_i$ and $\mu_i$ are the number and the chemical potential of species $i$, respectively, presented in the system, $i=$ Co or S. The two chemical potentials, $\mu_{Co}$ and $\mu_S$, are related via the Gibbs free energy of the bulk under the equilibrium condition, that is, $\mu_{Co} + 2\mu_S = g_{CoS_2}^{\mathrm{bulk}}$, where $g_{CoS_2}^{\mathrm{bulk}}$ denotes the Gibbs free energy per formula unit. Combination with equation (2.1), a surface free energy as a function of the chemical potential of S is obtained as

$$\gamma(T,p) = \frac{1}{A}[G^{\mathrm{surf}} - N_{Co}g_{CoS_2}^{\mathrm{bulk}} + (2N_{Co} - N_S)\mu_S]. \tag{2.2}$$

The term $\mu_S$ is restricted by the following conditions: (i) no Co metal or sulfur from CoS$_2$ decomposition, and (ii) no condensation of bulk sulfur on the surface. So the following relationship is obtained:

$$\Delta H_{f,CoS_2}^{\mathrm{bulk}}(T = 0, p = 0) < \mu_S - E_S^{\mathrm{bulk}} < 0. \tag{2.3}$$

The term $\Delta H_{f,CoS_2}^{\mathrm{bulk}}(T = 0, p = 0)$ is the low-temperature limit for the formation heat of CoS$_2$, while $E_S^{\mathrm{bulk}}$ is the total energy of S atom in the $\alpha$ phase of bulk sulfur.

# 3. Results and discussion

## 3.1. Bulk CoS$_2$

CoS$_2$ crystallizes in a rock-salt structure with space group symmetry of PA3 [29]. There are four formula units of CoS$_2$ in the face-centred cubic cell, as shown in figure 1. The Co atoms are situated at all corners and face centre positions, and the S$_2$ dimers are at the centre and midpoints of the twelve edges of the unit cell. Table 1 shows the comparison of the geometrically optimized lattice parameters with the experimental values, and the difference was small, indicating that the established model is acceptable.

## 3.2. CoS$_2$ surface models

The (111), (200), (210), (211) and (220) surfaces of CoS$_2$ were investigated, and each surface has different numbers of sections due to the different termination atoms. The (111) face is a hexagonal, and there are five different terminations on this surface, namely (111)-Co with three coordination atoms in the outer layer and (111)-S, (111)-2S, (111)-3S, (111)-4S with one, two, three or four layers of S atoms in the outer layer, shown in figure 2. The (200) surface is square and has three different terminations based on the outermost atoms species, which were (200)-Co with three coordination atoms in the outermost layer, (200)-S and (200)-2S with one or two S atoms in the outer layer, respectively, as is shown in figure 3. Figure 4 shows that six different terminations are on the (210) face, which is rectangular. There are two sections of (210)-S and (210)-S′, in which S atoms are 2-coordinated and 3-coordinated, respectively, when the outermost layer is one S-atom layer. Two sections of (210)-2S and (210)-2S′

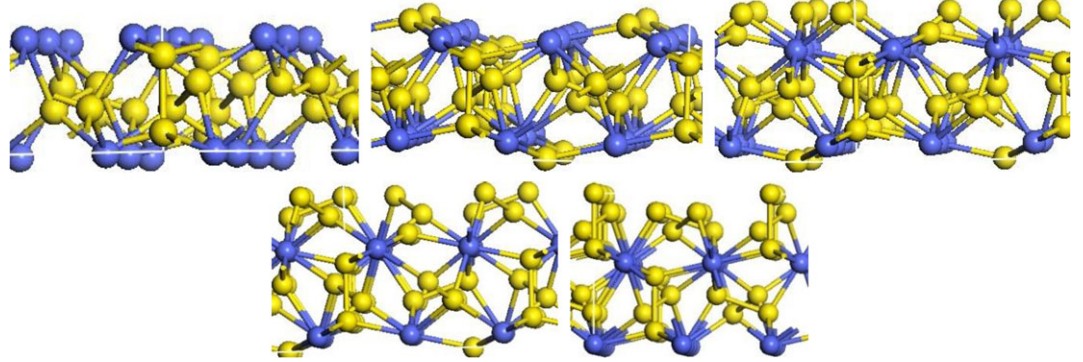

**Figure 2.** Different terminations of the CoS$_2$ (111) surface. Blue spheres indicate Co atoms and yellow spheres indicate S atoms.

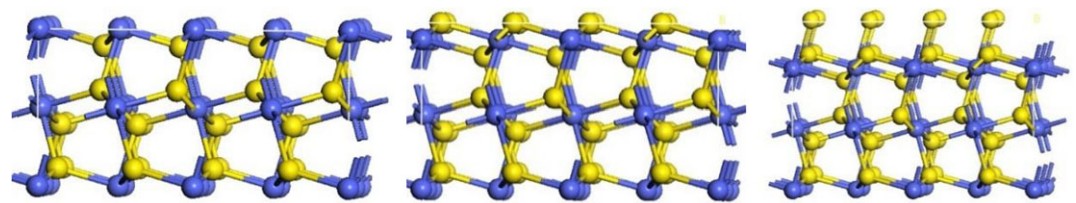

**Figure 3.** Different terminations of the CoS$_2$ (200) surface.

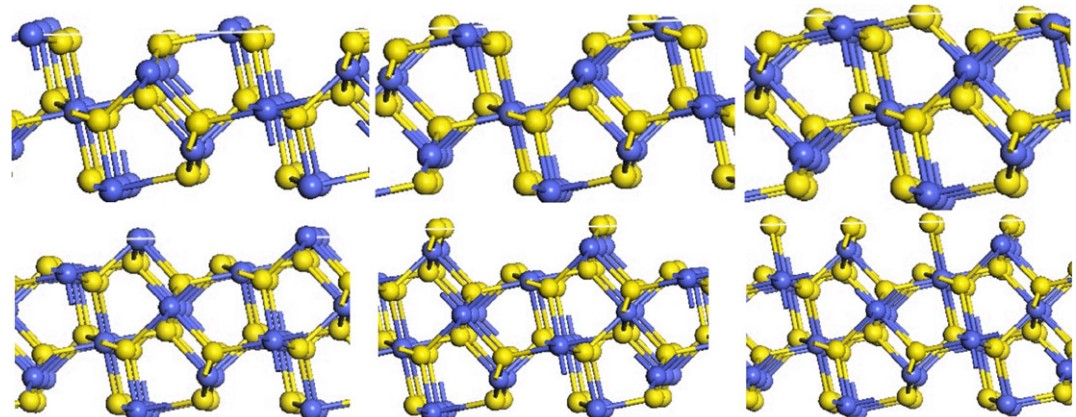

**Figure 4.** Different terminations of the CoS$_2$ (210) surface.

**Table 1.** Lattice parameter of CoS$_2$ crystal (nm).

|  | a | u |
|---|---|---|
| calculated value | 0.5515 | 0.0391 |
| experimental values [30,31] | 0.5524 | 0.0389 |

have the two S-atom layers on the outermost. S atoms in (210)-2S are 2-coordinated and 1-coordinated, while S atoms in (210)-2S′ are 2-coordinated and 3-coordinated. There are two different Co-terminated surfaces of (210)-Co and (210)-Co′, in which S atoms are both 2-coordinated. Figure 5 shows that the (211) face is also rectangular and has seven different terminations, which are (211)-3S, (211)-2S, (211)-S, (211)-CoS, (211)-2S′, (211)-S and (211)-Co. The (220) face is rectangular with three different terminations of (220)-S, (220)-2S and (220)-CoS, shown in figure 6.

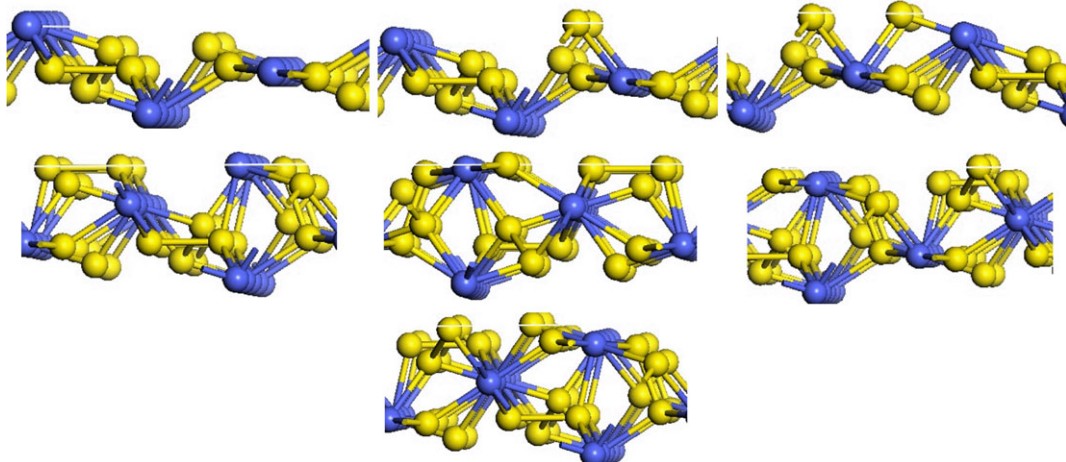

**Figure 5.** Different terminations of the CoS$_2$ (211) surface.

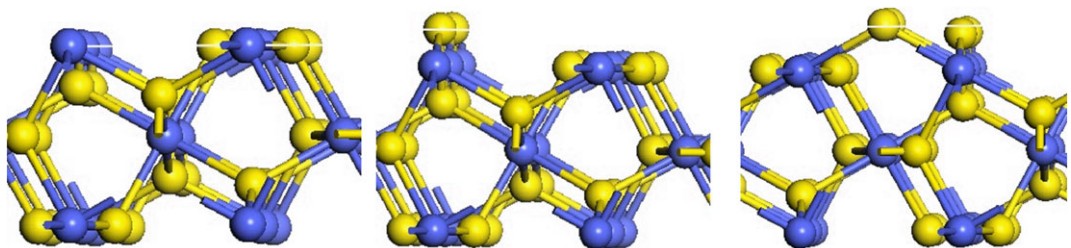

**Figure 6.** Different terminations of the CoS$_2$ (220) surface.

## 3.3. Surface energy

The calculated surface energy of different termination surfaces varying with S chemical potential ($\mu_S$) is displayed in figure 7. The vertical dashed lines represent the upper and lower bounds of S specified by equation (2.3). The lower limit labelled as S-poor condition is defined as the equilibrium state achieved by decomposition of CoS$_2$ into Co and S, and the upper limit is S-rich, which corresponds to the state of accumulation and deposition of gaseous sulfur on the surface. Since there is no coefficient including $\mu_S$ in equation (2.2), it is speculated that the surface free energy of the stoichiometric surfaces is independent of $\mu_S$, which is confirmed by the straight lines in the figure 7. These lines correspond to the surfaces of (111)-4S, (200)-2S, (210)-2S, (211)-3S, (220)-2S and (210)-2S. The order of surface energy increase is $E_{\text{Surf}}$ (220)-2S < $E_{\text{Surf}}$ (211)-3S < $E_{\text{Surf}}$ (111)-4S < $E_{\text{Surf}}$ (210)-2S < $E_{\text{Surf}}$ (200)-2S, which is opposite to that of thermodynamical stability. Moreover, the stoichiometric surface (210)-2S′ is more stable than (210)-2S.

For non-stoichiometric surfaces, the surface energy is a linear function of $\mu_S$ from $\Delta\mu_S = \mu_S - E_S^{bulk}$, corresponding to the oblique lines in the figure 7. Obviously, (220)-2S has the lowest surface energy under the S-rich condition, while the non-stoichiometric surfaces are more stable with the chemical potential lower than S-poor. The Co-terminated surfaces have higher surface energy, especially for the (200)-Co and (111)-Co. The surface energy of (111)-S surface varies greatly with $\Delta\mu_S$. (211)-2S has the lower surface energy when $\Delta\mu_S$ is larger than −1.0 eV. (211)-S is the most stable surface with $\Delta\mu_S$ from −1.0 to −1.55 eV, while the (211)-Co and (111)-S will have the lowest surface energy when $\Delta\mu_S$ is less than −1.55 eV.

## 3.4. Electronic structure

We calculated the electronic structures of the stoichiometric surfaces (111)-4S, (200)-2S, (210)-2S, (211)-3S and (220)-2S. Figures 8, 9 and 10 show the results of Fermi energy of the selected surfaces slabs, the total density of states and the band structures, respectively.

As seen in figure 8, the highest surface energy (200)-2S surface has the highest Fermi energy, followed by (111)-4S surface, and the Fermi energy of (211)-3S surface is slightly higher than that of (210)-2S surface, which is the lowest. The thermodynamic stability of (200)-2S surface is supposed to be stable

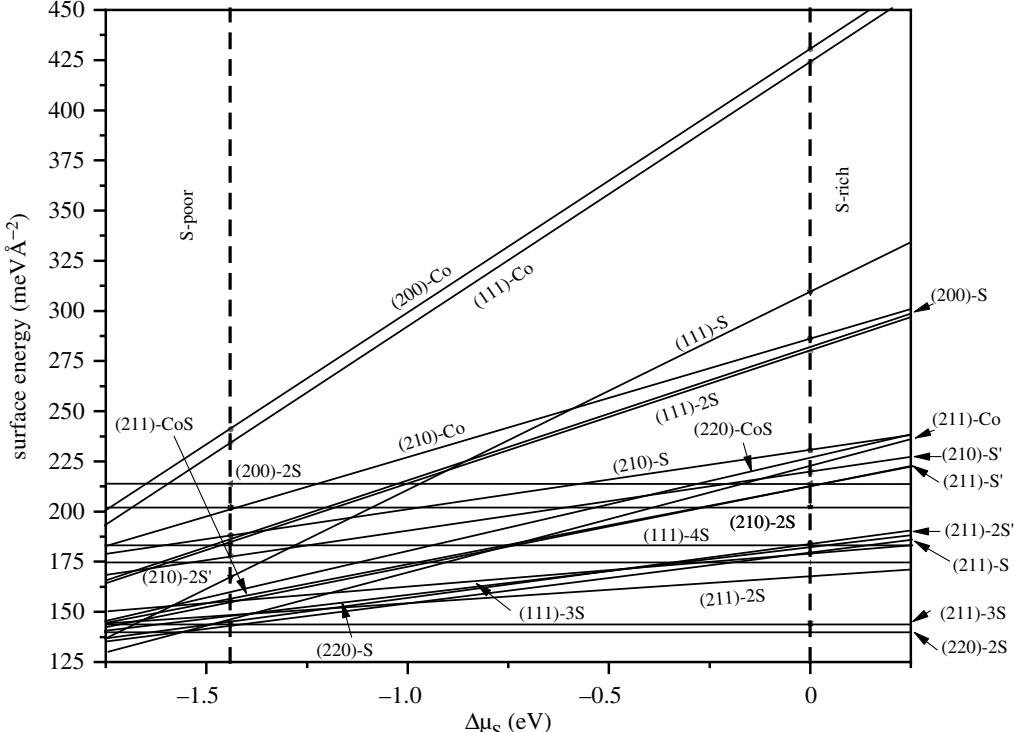

**Figure 7.** Calculated surface free energies of various CoS$_2$ surfaces as functions of S chemical potential.

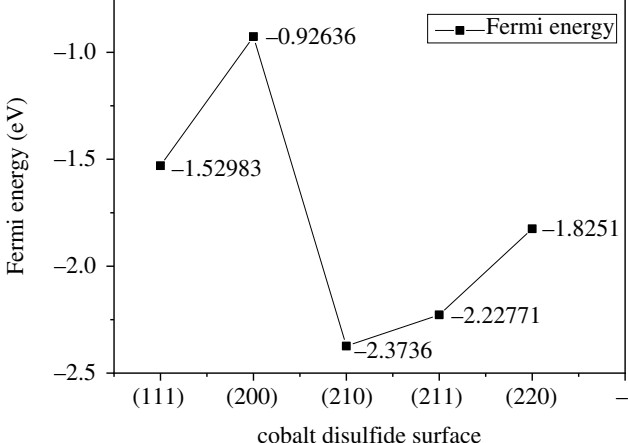

**Figure 8.** Fermi energy of cobalt disulfide surfaces.

due to its high surface energy, leading to the fast growth rate of this surface. However, the high Fermi energy of this surface indicates that its front valence electrons are active, resulting in transferring electrons with the lowest non-occupied band of the matched reactant easily. There may be 'active points' that bond with ions, molecules or crystal growth units in solution. Once adsorption occurs, the surface energy of the surface will be changed, resulting in the change of the growth rate of the surface, and then the crystal morphology. On the other hand, the other surface slabs have the lower Fermi energy, implying that their front valence electrons are less active and have fewer 'active points'. Especially for the (211)-3S with a very low surface energy, this surface is relatively stable in terms of thermodynamics and dynamics of electronic structure.

Figure 9 shows that there are some differences in the energy distribution range, peak number and peak value on (111)-4S, (200)-2S, (210)-2S, (211)-3S and (220)-2S surface slabs. This is more obvious in the band structures of figure 10. As shown in figure 9, the peak values on (111)-4S and (211)-3S planes are close, while those on (200)-2S and (210)-2S planes are similar and lower than the former. The

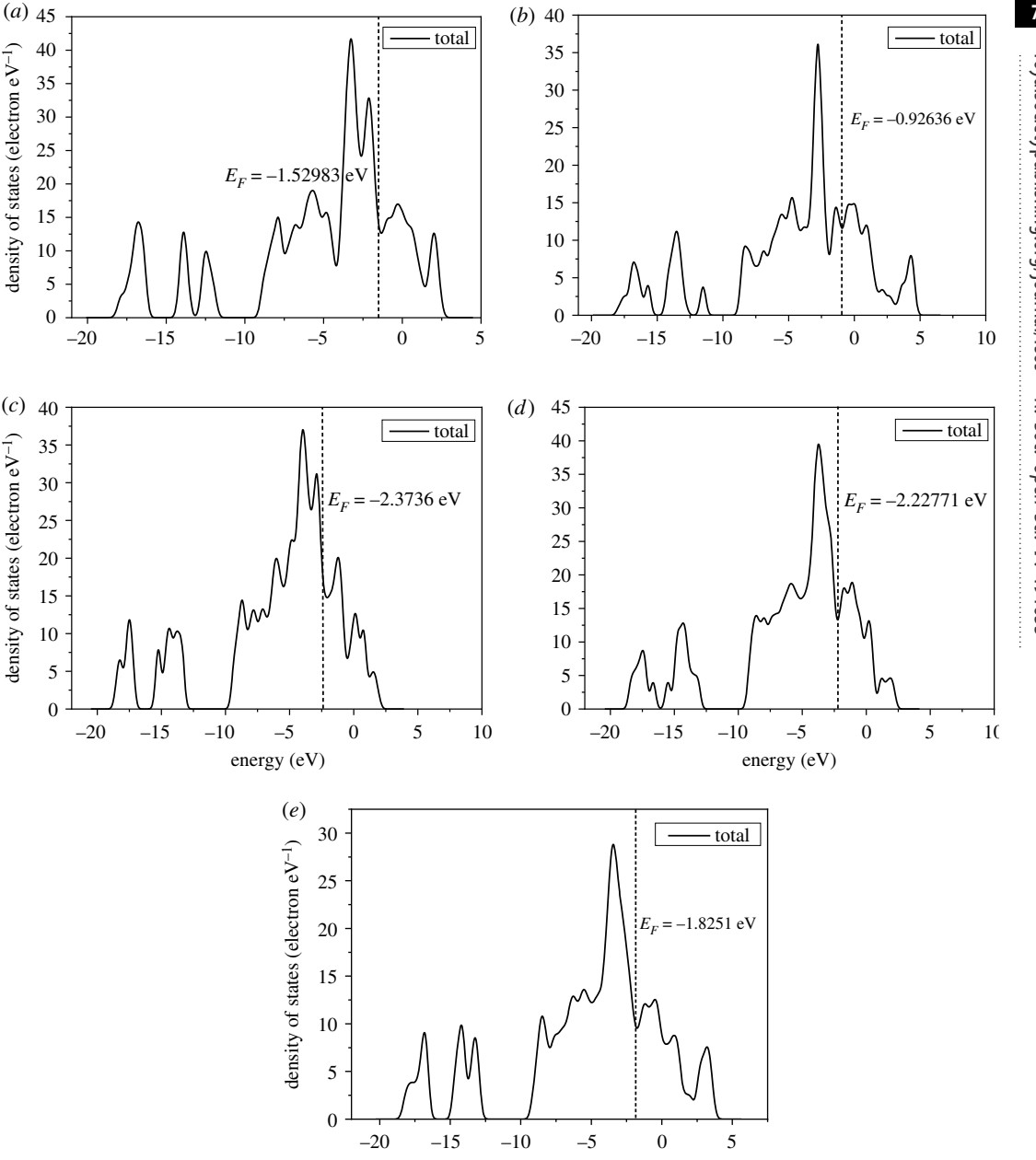

**Figure 9.** Density of states of cobalt disulfide surface slab models: (*a*) 111-4S, (*b*) 200-2S, (*c*) 210-2S, (*d*) 211-2S and (*e*) 220-2S.

sub-peak on (111)-4S at Fermi level is wider than that of other facets, and the difference between the sub-peak and the peak at the lower energy region on (210)-2S is smaller than that of other facets, indicating that the number of energy bands with higher energy on these two surfaces is larger. On the other hand, there is no secondary peak near the Fermi energy for (211)-3S, which indicates that the thermodynamic properties of (211) surface are stable. The band structures (figure 10) show that the band fluctuation near Fermi level at (200)-2S surface is large, which indicates that the effective mass of the electron is small, the degree of non-locality is large and the atomic orbital expansion of the band is strong. Figure 10 also shows that there are energy gaps on (211)-3S and (220)-2S slabs, which are 0.035 and 0.128 eV, respectively, indicating that their inner electrons are stable and (220)-2S surface is more stable than (211)-3S. Therefore, we can say that the dynamic stability of the electronic structure of (220)-2S in the system is better than those of other surfaces.

From the band structure diagram, the Fermi levels on (111)-4S, (200)-2S and (210)-2S surfaces pass through the valence band, showing metallicity. Especially for (111)-4S, its Fermi level passes through the valence bands of spin-up and spin-down band structure, thus showing complete metallicity. The

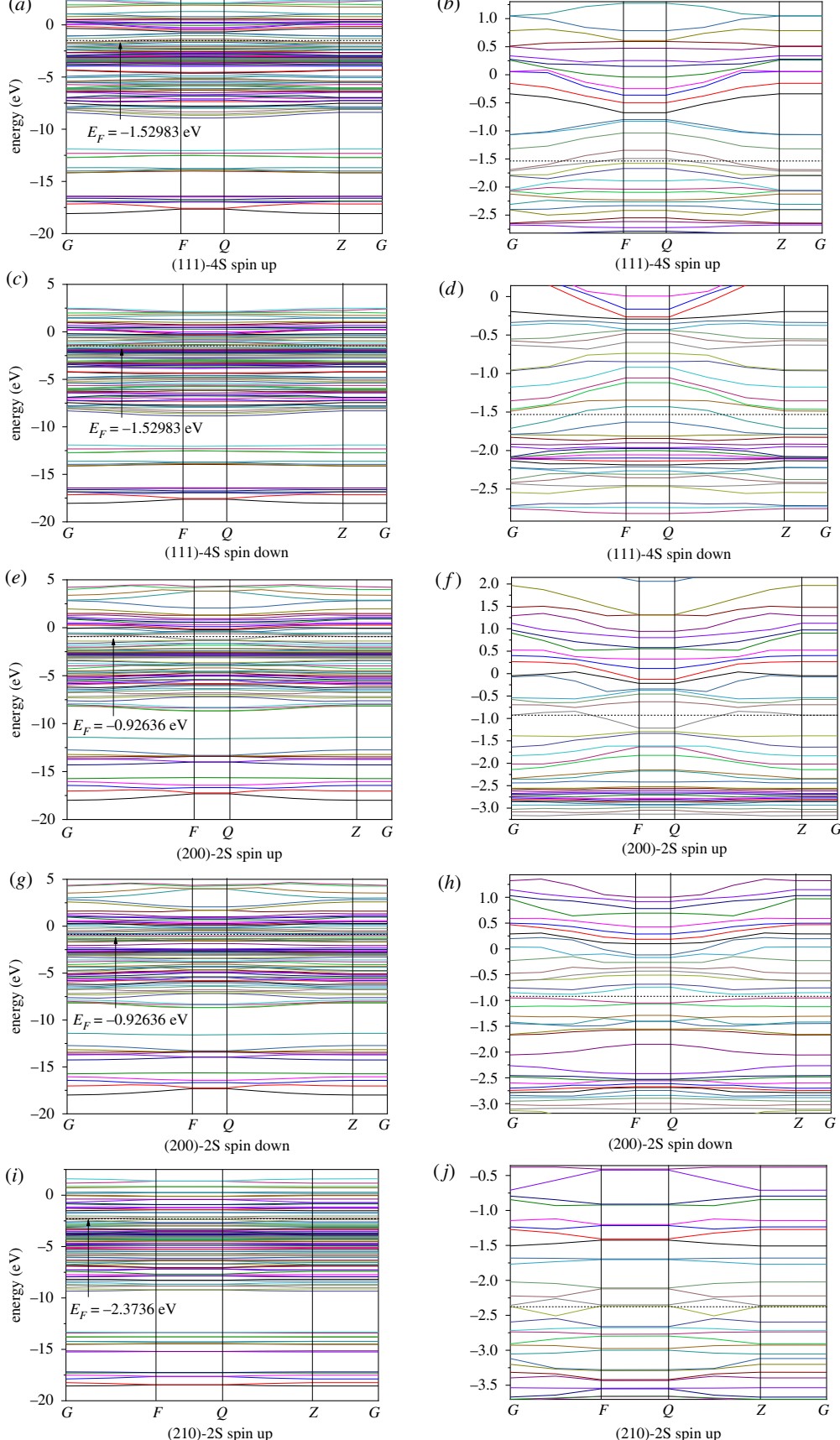

**Figure 10.** Band structures of cobalt disulfide surface slab models.

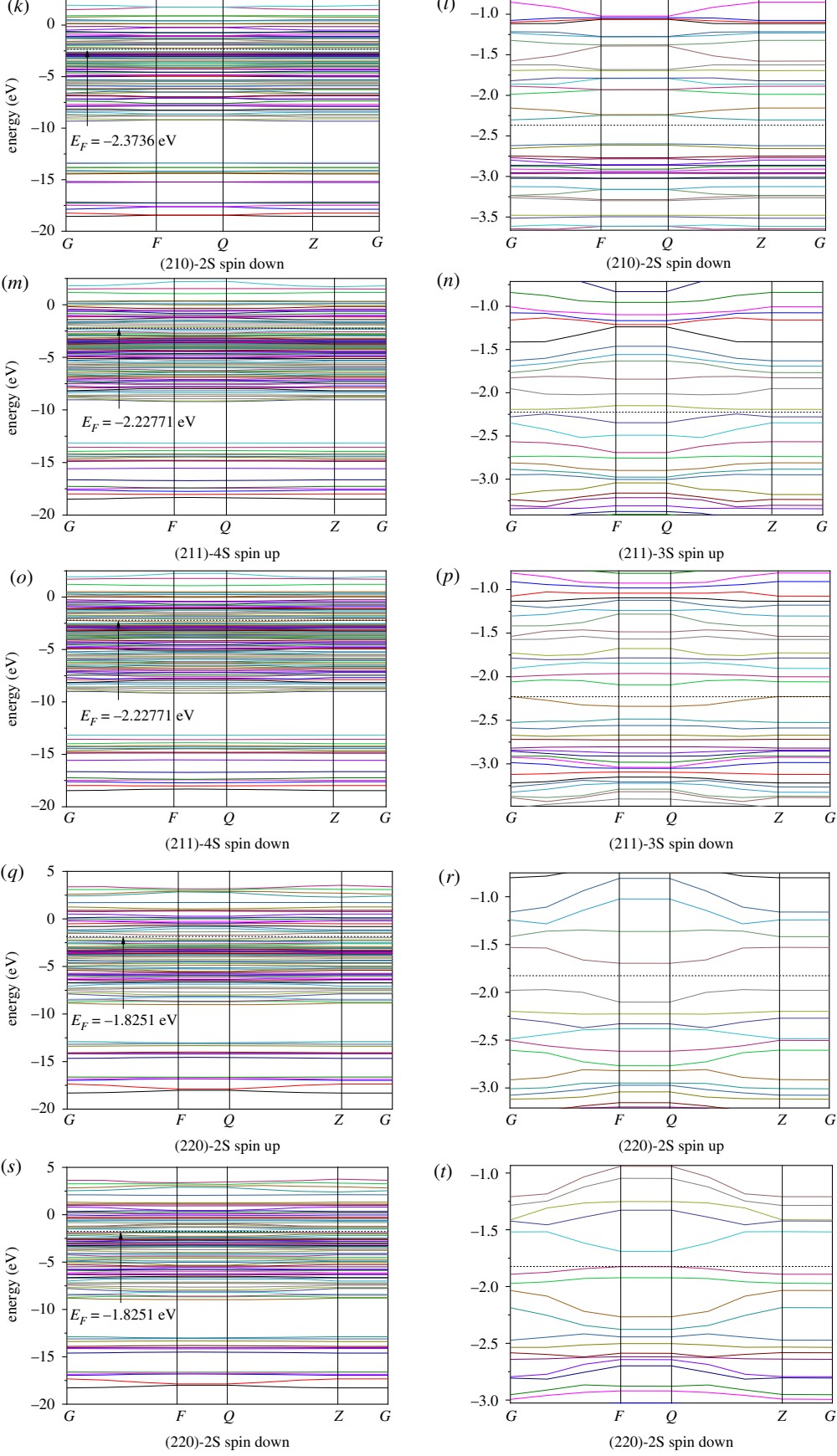

**Figure 10.** (Continued.)

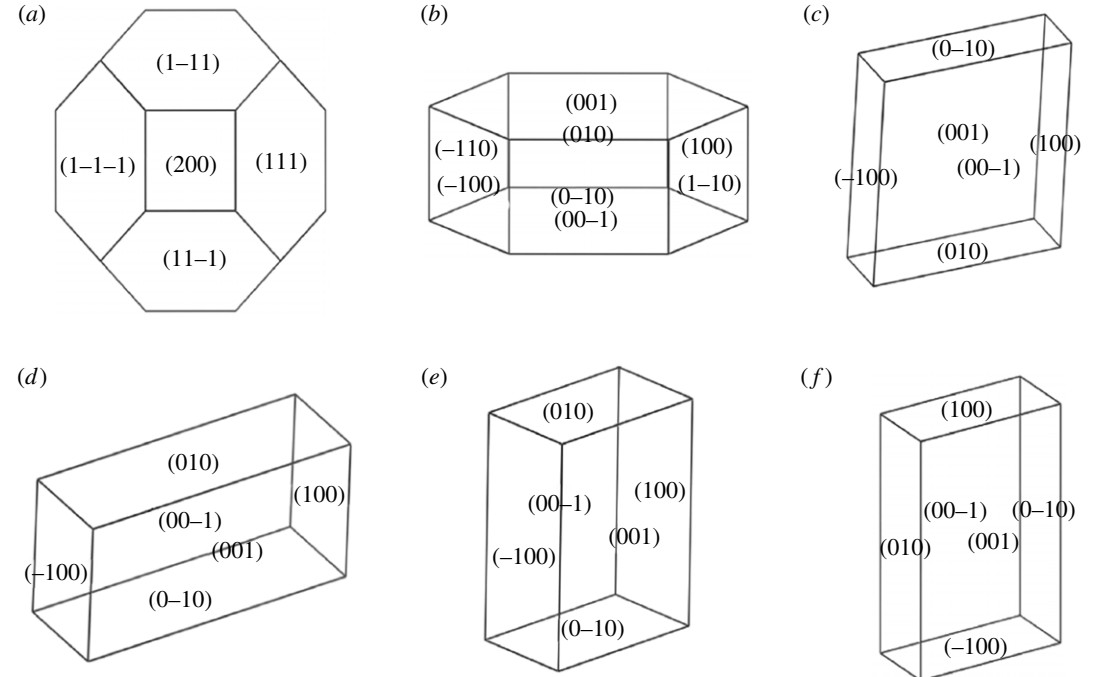

**Figure 11.** Crystal growth habits of CoS$_2$ and its surface slab models calculated by BFDH method. (*a–f*) Correspond to CoS$_2$ bulk cell, (111), (200), (210), (211) and (220), respectively.

Fermi levels on (211)-3S and (220)-2S surfaces are between valence bands and conduction bands, which make them exhibit semiconductor property. However, the Fermi level of (200)-2S and (210)-2S only passes through the valence band of spin-up band structure, but does not pass through the valence band of spin-down band structure, showing semi-metallicity and high spin polarization.

## 3.5. Morphology prediction by Bravais–Friedel–Donnay–Harker method

In order to learn about the importance of the surfaces on growth morphology, Bravais–Friedel–Donnay–Harker (BFDH) method was employed to simulate and predict the macroscopic morphology of CoS$_2$. As is shown in figure 11*a*, (200) and (111) are the important surfaces with the importance order of (111) > (200). These results successfully predict the exposed faces (111) and (200), and (111) is more thermodynamically stable than (200), which is consistent with the results obtained from the above surface energy calculation. Moreover, it can be inferred from the crystal morphology that CoS$_2$ crystal will tend to octahedron if (111) is the main exposed surface, the crystal will tend to cube if (200) is the main exposed surface, and it will tend to quasi-spherical if (200) and (111) are both exposed surfaces.

The growth habit of the selected surface slab models was predicted using the same method, and the results are shown in figure 11*b–f*.

The (111) surface tends to grow into hexagonal or hexagonal plate-like crystals, (200), (211) and (220) surfaces tend to grow into quadrangular -like crystals, while (210) tends to grow into plate-like crystals. Therefore, it can be inferred that there may be spherical, cubic, octahedral, prismatic and plate-like CoS$_2$ crystals in the growth of CoS$_2$. CoS$_2$ octahedron can be prepared by hydrothermal synthesis according to our previous results [8,32]. Other CoS$_2$ crystals with different morphologies were also confirmed by experiments, the spherical by Wang's group [33], flakes obtained by a simple chemical bath deposition (CBD) method [34] and prisms by prismatic Co-precursors [35].

## 4. Conclusion

The various terminations of (111), (200), (210), (211) and (220) surfaces using the first-principle and the slab technique were investigated. The stability on thermodynamic of the various terminations in the arbitrary sulfur environment was examined by surface energy. The stoichiometric surfaces (220)-2S

and (211)-3S are more stable under S-rich condition, while (210)-Co and (111)-S become more stable surfaces under S-poor conditions.

The electronic structure of five stoichiometric surfaces were calculated, and the results show that the front valence electrons of (200)-2S with the highest Fermi energy are active, indicating that there may be 'active points' on this surface and easy to bond with ions, molecules or crystal growth units in solution. The Fermi energies of (220)-2S and (211)-3S are low, and their inner electrons are stable, showing that the dynamic stability of electronic structure is good. Moreover, the high energy occupying bands are less, which again shows that they have good thermodynamic stability when they are the main exposed surfaces. The spin polarization of (200)-2S and (210)-2S surfaces is high. The calculated results by BFDH method showed that the important crystal planes of $CoS_2$ crystal growth were (111) and (200). The macroscopic morphology of $CoS_2$ crystal may be spherical, cubic, octahedral, prismatic and plate, which is confirmed by experiments.

Permission to carry out fieldwork. Our data sources do not include fieldwork.

Ethics. Our investigation was carried out in full accordance with the ethical guidelines of our research institution and in compliance with Chinese legislation.

Data accessibility. Data supporting this study are available from the Dryad Digital Repository: https://doi.org/10.5061/dryad.wh70rxwhv [36].

Authors' contributions. Y.Z. designed and directed the calculation and amended the manuscript; C.W. wrote the manuscript and submitted the manuscript; F.G. cooperated on this project; Z.X. calculated the surface energy and electronic properties; P.Z. provided technical support for calculations. J.W. administrated the project. All authors gave final approval for publication.

Competing interests. We declare we have no competing interests.

Funding. This work was financially supported by National Natural Science Foundation of China (grant nos. 51974031 and 51774044).

Acknowledgements. We thank Qing-fen Meng and Qi-bing Wu for their guidance in calculation and helpful suggestions.

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
