## [Reviewer comments · Royal Society Open Science]

Review History

RSOS-191653.R0 (Original submission)

Review form: Reviewer 1

Is the manuscript scientifically sound in its present form?

No

Are the interpretations and conclusions justified by the results?

No

Is the language acceptable?

No

Do you have any ethical concerns with this paper?

No

Have you any concerns about statistical analyses in this paper?

No

Recommendation?

Major revision is needed (please make suggestions in comments)

Comments to the Author(s)

Here are some important points to discuss:

- 1) Why CoS₂, does your structure is not Co₉S₈?
- 2) There is no motivation, authors just mentioned on the introduction that CoS₂ can be used on many technological applications. But, doesn't present any experimental evidence.
- 3) Authors, should focus on finding any catalytic performance, at least for hydrogen. This is an easy calculation to achieve using DFT.
- 4) What is the point on putting a simulated XRD plot? There is no description behind.
- 5) Both the DOS and BS are vague. It seems that indeed they don't have a periodic system. Thus, both sets of data does not represent bulk. And on their conclusions, surface configurations are mostly presented for the bulk phase.
- 6) Authors, should include citations on CoS₂ evidence. Or Co₉S₈ experimental evidence (i.e. Catal. Sci. Technol., 2012, 2, 164–178).
- 7) Overall the English grammar should be improved.
- 8) Statements such: "There may be "active points" that bond with ions", are vague if there is lack of data about binding chemical elements onto those mentioned sites.

Review form: Reviewer 2

Is the manuscript scientifically sound in its present form?

Yes

Are the interpretations and conclusions justified by the results?

Yes

Is the language acceptable?

Yes

Do you have any ethical concerns with this paper?

Yes

Have you any concerns about statistical analyses in this paper?

No

Recommendation?

Major revision is needed (please make suggestions in comments)

Comments to the Author(s)

In this work, the surface energy of CoS₂ for different crystallographic plane have been calculated by DFT calculation. The results are interesting and help understanding the morphology of CoS₂ particles. I have several points for the authors' clarification.

- 1) In the section 'computational approach (P.3, line 53-55), the authors say "The slab thickness was between 6 and 16 atomic layers (14-24 atoms) based on the restriction of computational ability. All the atoms of the slab were relaxed with the five bottom layers fixed to their bulk values." For the geometry model with 16 atomic layers, it is reasonable that the structure relaxation are done with five bottom layers fixed to their bulk values. However, for the structural model of total 6 atomic layers, it is not a good approximation with five bottom layers fixed to their bulk values.
- 2) The surface structural stability and crystal morphology is sensitive to total free energy. The authors do not include the contribution from atomic vibration or phonon part in the

calculation. I suggest the authors add some discussion on the possible effect from atomic vibration at least.

Decision letter (RSOS-191653.R0)

24-Feb-2020

Dear Professor Zhu:

Title: Calculation on surface energy and electronic properties of CoS₂
Manuscript ID: RSOS-191653

The editor assigned to your manuscript has now received comments from reviewers. We would like you to revise your paper in accordance with the referee and Subject Editor suggestions which can be found below (not including confidential reports to the Editor). Please note this decision does not guarantee eventual acceptance.

Please submit your revised paper before 18-Mar-2020. Please note that the revision deadline will expire at 00.00am on this date. If we do not hear from you within this time then it will be assumed that the paper has been withdrawn. In exceptional circumstances, extensions may be possible if agreed with the Editorial Office in advance. We do not allow multiple rounds of revision so we urge you to make every effort to fully address all of the comments at this stage. If deemed necessary by the Editors, your manuscript will be sent back to one or more of the original reviewers for assessment. If the original reviewers are not available we may invite new reviewers.

On behalf of the Subject Editor Professor Anthony Stace and the Associate Editor Professor Kim Jelfs.

RSC Associate Editor:

Comments to the Author:

Please carefully answer and improve the manuscript based on all the reviewers points. Reviewer 1 has also asked that Figure 11 be improved.

RSC Subject Editor:

Comments to the Author:

(There are no comments.)

Reviewers' Comments to Author:

Reviewer: 1

Comments to the Author(s)

Here are some important points to discuss:

- 1) Why CoS₂, does your structure is not Co₉S₈?
- 2) There is no motivation, authors just mentioned on the introduction that CoS₂ can be used on many technological applications. But, doesn't present any experimental evidence.
- 3) Authors, should focus on finding any catalytic performance, at least for hydrogen. This is an easy calculation to achieve using DFT.
- 4) What is the point on putting a simulated XRD plot? There is no description behind.
- 5) Both the DOS and BS are vague. It seems that indeed they don't have a periodic system. Thus, both sets of data does not represent bulk. And on their conclusions, surface configurations are mostly presented for the bulk phase.
- 6) Authors, should include citations on CoS₂ evidence. Or Co₉S₈ experimental evidence (i.e. Catal. Sci. Technol., 2012, 2, 164–178).
- 7) Overall the English grammar should be improved.
- 8) Statements such: "There may be "active points" that bond with ions", are vague if there is lack of data about binding chemical elements onto those mentioned sites.

Reviewer: 2

Comments to the Author(s)

In this work, the surface energy of CoS₂ for different crystallographic plane have been calculated by DFT calculation. The results are interesting and help understanding the morphology of CoS₂ particles. I have several points for the authors' clarification.

- 1) In the section 'computational approach (P.3, line 53-55), the authors say "The slab thickness was between 6 and 16 atomic layers (14-24 atoms) based on the restriction of computational ability. All the atoms of the slab were relaxed with the five bottom layers fixed to their bulk values." For the geometry model with 16 atomic layers, it is reasonable that the structure relaxation are done with five bottom layers fixed to their bulk values. However, for the structural model of total 6 atomic layers, it is not a good approximation with five bottom layers fixed to their bulk values.
- 2) The surface structural stability and crystal morphology is sensitive to total free energy. The authors do not include the contribution from atomic vibration or phonon part in the

calculation. I suggest the authors add some discussion on the possible effect from atomic vibration at least.

Author's Response to Decision Letter for (RSOS-191653.R0)

See Appendix A.

RSOS-191653.R1 (Revision)

Review form: Reviewer 1

Is the manuscript scientifically sound in its present form?

Yes

Are the interpretations and conclusions justified by the results?

Yes

Is the language acceptable?

Yes

Do you have any ethical concerns with this paper?

No

Have you any concerns about statistical analyses in this paper?

No

Recommendation?

Accept as is

Comments to the Author(s)

Dear Authors;

I am pleased to be able to provide you with some feedback and insights about your data and overall manuscript as provided on the original submission file to Royal Society Open Science. I was able to see that you follow all advise provided by my self and that you were able to improve the quality of your manuscript.

Great manuscript, good science!

Review form: Reviewer 2

Is the manuscript scientifically sound in its present form?

Yes

Are the interpretations and conclusions justified by the results?

Yes

Is the language acceptable?

Yes

Do you have any ethical concerns with this paper?

No

Have you any concerns about statistical analyses in this paper?

No

Recommendation?

Accept with minor revision (please list in comments)

Comments to the Author(s)

I am happy to see improved version of the manuscript. I have still question of my point 1 in previous comments.

In the part "computation approach" (P.8, third paragraph, line 40-44), it states "The slab thickness was between 6 and 16 atomic layers (14-24 atoms) based on the restriction of computational ability. All the atoms of the slab were relaxed with the eight bottom layers fixed to their bulk values. The lattice constants were fixed at bulk optimized conditions."

My question, is, for the slab with a thickness of 6 atomic layers, how to relax the slab with eight bottom layers fixed to their bulk values? Please clarify it.

Decision letter (RSOS-191653.R1)

15-Apr-2020

Dear Professor Zhu:

Title: Calculation on surface energy and electronic properties of CoS₂
Manuscript ID: RSOS-191653.R1

Thank you for submitting the above manuscript to Royal Society Open Science. On behalf of the Editors and the Royal Society of Chemistry, I am pleased to inform you that your manuscript will be accepted for publication in Royal Society Open Science subject to minor revision in accordance with the referee suggestions. Please find the reviewers' comments at the end of this email.

The reviewers and handling editors have recommended publication, but also suggest some minor revisions to your manuscript. Therefore, I invite you to respond to the comments and revise your manuscript.

Because the schedule for publication is very tight, it is a condition of publication that you submit the revised version of your manuscript before 24-Apr-2020. Please note that the revision deadline will expire at 00.00am on this date. If you do not think you will be able to meet this date please let me know immediately.

Kind regards,

Dr Laura Smith
Publishing Editor, Journals

On behalf of the Subject Editor Professor Anthony Stace and the Associate Editor Professor Kim Jelfs.

RSC Associate Editor:
Comments to the Author:
Please ensure that you complete the final requests of the reviewer.

RSC Subject Editor:
Comments to the Author:
(There are no comments.)

Reviewer comments to Author:
Reviewer: 2

Comments to the Author(s)

I am happy to see improved version of the manuscript. I have still question of my point 1 in previous comments.

In the part "computation approach" (P.8, third paragraph, line 40-44), it states "The slab thickness was between 6 and 16 atomic layers (14-24 atoms) based on the restriction of computational ability. All the atoms of the slab were relaxed with the eight bottom layers fixed to their bulk values. The lattice constants were fixed at bulk optimized conditions."

My question, is, for the slab with a thickness of 6 atomic layers, how to relax the slab with eight bottom layers fixed to their bulk values? Please clarify it.

Reviewer: 1

Comments to the Author(s)
Dear Authors;

I am pleased to be able to provide you with some feedback and insights about your data and overall manuscript as provided on the original submission file to Royal Society Open Science. I was able to see that you follow all advice provided by my self and that you were able to improve the quality of your manuscript.

Great manuscript, good science!

Author's Response to Decision Letter for (RSOS-191653.R1)

See Appendix B.

RSOS-191653.R2 (Revision)

Review form: Reviewer 2

Is the manuscript scientifically sound in its present form?

Yes

Are the interpretations and conclusions justified by the results?

Yes

Is the language acceptable?

Yes

Do you have any ethical concerns with this paper?

No

Have you any concerns about statistical analyses in this paper?

No

Recommendation?

Accept as is

Comments to the Author(s)

I have no concern on this new version of the paper. So I am happy to recommend it.

Decision letter (RSOS-191653.R2)

Dear Professor Zhu:

Title: Calculation on surface energy and electronic properties of CoS₂
Manuscript ID: RSOS-191653.R2

It is a pleasure to accept your manuscript in its current form for publication in Royal Society Open Science. The chemistry content of Royal Society Open Science is published in collaboration with the Royal Society of Chemistry.

On behalf of the Subject Editor Professor Anthony Stace and the Associate Editor Professor Kim Jelfs.

RSC Associate Editor:
Comments to the Author:

(There are no comments.)

RSC Subject Editor:

Comments to the Author:

(There are no comments.)

Reviewer(s)' Comments to Author:

Reviewer: 2

Comments to the Author(s)

I have no concern on this new version of the paper. So I am happy to recommend it.

Appendix A

Dear editors and reviewers:

Thank you for your affirmation of our article. The following is a reply one by one.

Reviewer: 1

Comments to the Author(s)

Here are some important points to discuss:

1) Why CoS₂, does your structure is not Co₉S₈?

Cobalt disulfide (CoS₂) is one of the idea cathode materials in thermal batteries with both high power and high energy output capacity, and Co₉S₈ is considered to be an intermediate product of CoS₂ discharge. Since our group's main research goal is to improve the performance of thermal batteries at present, thus, we chose CoS₂ as our research subject.

2) There is no motivation, authors just mentioned on the introduction that CoS₂ can be used on many technological applications. But, doesn't present any experimental evidence.

The introduction has been modified according to the reviewer's comments.

3) Authors, should focus on finding any catalytic performance, at least for hydrogen. This is an easy calculation to achieve using DFT.

Recently, our group's main research content is to improve the performance of thermal batteries, including the preparation of high-performance electrode materials, such as CoS₂ studied in this paper. Thank you for the suggestion, and the catalytic performance of sulfides is very interesting and we will do some relevant research in the future.

4) What is the point on putting a simulated XRD plot? There is no description behind.

The XRD plot (Fig. 1) was deleted from the text.

5) Both the DOS and BS are vague. It seems that indeed they don't have a periodic system. Thus, both sets of data does not represent bulk. And on their conclusions, surface configurations are mostly presented for the bulk phase.

Both the DOS and BS (Figs. 10 and 11) have been improved, and they showed the surface configurations of the cobalt disulfide surface slab models. The macroscopic morphology of CoS₂ was simulated and predicted by BFDH method in Section 3.5, and the effects of the surfaces on growth morphology were discussed. There results showed that the (111) and (200) planes played an important role in the CoS₂ crystal growth, which were consistent with the calculation results in Sections 3.3 and 3.4. The corresponding statements in the conclusions have been modified.

6) Authors, should include citations on CoS₂ evidence. Or Co₉S₈ experimental evidence (i.e. Catal. Sci. Technol., 2012, 2, 164–178).

The introduction has been modified, and the suggested literature has been cited.

7) Overall the English grammar should be improved.

We have tried our best to improve the English.

8) Statements such: "There may be "active points" that bond with ions", are vague if there is lack of data about binding chemical elements onto those mentioned sites.

This inference was based on the calculation results of surface energy and Fermi energy in Sections 3.3 and 3.4 in this study, and it was supported by the calculation about the adsorption capacity of water molecule on the various surfaces by DFT and CASTEP procedures (not shown in the text). These results are shown in the following table and figure, which show that the adsorption capacity of water molecule on the (200) surface is the strongest.

Total Energy and final adsorption energy of water molecules adsorbed on different adsorption positions on each crystal surface

Surfaces (hkl)	Mode	$E_{\text{sys}}(\text{eV})$	$E_{\text{slab}}(\text{eV})$	$E_{\text{ads}}(\text{kJ/mol})$
(111)	Co-top	-11590.60	-11122.16	35.58
	S-top	-11590.48		
	Bridge	-11590.87		
	Hollow	-11590.71		
(200)	Co-top	-9511.50	-9038.49	72.11
	S-top	-9511.06		
	Bridge	-9511.47		
	Hollow	-9511.34		
(210)	Co-top	-12142.56	-11673.81	35.58
	S-top	-12142.08		
	Bridge	-12142.39		
	Hollow	-12142.41		
(211)	Co-top	-9223.51	-8754.71	40.38
	S-top	-9223.36		

	Bridge	-9223.35		
	Hollow	-9223.36		
	Co-top	-9506.16		
	S-top	-9505.98		
(220)			-9037.37	39.42
	Bridge	-9506.07		
	Hollow	-9506.85		

Figure. Stable adsorption configuration of water molecules on each crystal surface of CoS₂

Reviewer: 2

Comments to the Author(s)

In this work, the surface energy of CoS₂ for different crystallographic plane have been calculated by DFT calculation. The results are interesting and help understanding the morphology of CoS₂ particles. I have several points for the authors' clarification.

Thank you for your affirmation of our work. The responses to your points are as follows:

1) In the section 'computational approach (P.3, line 53-55), the authors say "The slab thickness was between 6 and 16 atomic layers (14-24 atoms) based on the restriction of computational ability. All the atoms of the slab were relaxed with the five bottom layers fixed to their bulk values." For the geometry model with 16 atomic layers, it is reasonable that the structure relaxation are done with five bottom layers fixed to their bulk values. However, for the structural model of total 6 atomic layers, it is not a good approximation with five bottom layers fixed to their bulk values.

Thank you for questioning. We have corrected the corresponding mistake, and it was eight bottom layers, just equal to the atomic number of bulk CoS₂.

2) The surface structural stability and crystal morphology is sensitive to total free energy. The authors do not include the contribution from atomic vibration or phonon part in the calculation. I suggest the authors add some discussion on the possible effect from atomic vibration at least.

We did not consider the effects of atomic vibration or phonon part in this study, and the stability of the various surfaces were investigated by associating DFT results with thermodynamic concepts, mainly from a macro perspective. The effects of atomic vibration or phonon part on the free energy are being carried out, hoping to systematically analyze the impact of various parts.

Appendix B

Dear editors and reviewers:

Thank you for your affirmation of our article. The following is a reply one by one.

Reviewer: 2

Comments to the Author(s)

I am happy to see improved version of the manuscript. I have still question of my point 1 in previous comments.

In the part "computation approach" (P.8, third paragraph, line 40-44), it states "The slab thickness was between 6 and 16 atomic layers (14-24 atoms) based on the restriction of computational ability. All the atoms of the slab were relaxed with the eight bottom layers fixed to their bulk values. The lattice constants were fixed at bulk optimized conditions."

My question, is, for the slab with a thickness of 6 atomic layers, how to relax the slab with eight bottom layers fixed to their bulk values? Please clarify it.

Thank you very much for your question. We are very sorry for serious expression errors. In the calculation process, we selected 10-16 atomic layers, of which the bottom 8 atoms are fixed. The corresponding content in the article has been revised.

Reviewer: 1

Comments to the Author(s)

Dear Authors;

I am pleased to be able to provide you with some feedback and insights about your data and overall manuscript as provided on the original submission file to Royal Society Open Science. I was able to see that you follow all advice provided by myself and that you were able to improve the quality of your manuscript.

Great manuscript, good science!